# Colon Cancer Microbiome Landscaping: Differences in Right- and Left-Sided Colon Cancer and a Tumor Microbiome-Ileal Microbiome Association

**DOI:** 10.3390/ijms24043265

**Published:** 2023-02-07

**Authors:** Barbara Kneis, Stefan Wirtz, Klaus Weber, Axel Denz, Matthias Gittler, Carol Geppert, Maximilian Brunner, Christian Krautz, Alexander Reinhard Siebenhüner, Robert Schierwagen, Olaf Tyc, Abbas Agaimy, Robert Grützmann, Jonel Trebicka, Stephan Kersting, Melanie Langheinrich

**Affiliations:** 1Department of Nephrology, University Hospital Erlangen, 91054 Erlangen, Germany; 2Department of Surgery, University Hospital Erlangen, 91054 Erlangen, Germany; 3Department of Internal Medicine I, University Hospital Erlangen, 91054 Erlangen, Germany; 4Institute of Pathology, University Hospital Erlangen, 91054 Erlangen, Germany; 5Department of Gastroenterology and Hepatology, University Hospital Zurich and University Zurich, 8091 Zürich, Switzerland; 6Department of Internal Medicine B, University Hospital Münster, 48149 Münster, Germany; 7Department of Internal Medicine I, University Clinic Frankfurt, 60590 Frankfurt, Germany; 8Department of Surgery, University Hospital Greifswald, 17475 Greifswald, Germany

**Keywords:** microbiome, colon cancer, right-sided colon cancer, left-sided colon cancer, tumor microbiome, gut microbiome

## Abstract

In the current era of precision oncology, it is widely acknowledged that CRC is a heterogeneous disease entity. Tumor location (right- or left-sided colon cancer or rectal cancer) is a crucial factor in determining disease progression as well as prognosis and influences disease management. In the last decade, numerous works have reported that the microbiome is an important element of CRC carcinogenesis, progression and therapy response. Owing to the heterogeneous nature of microbiomes, the findings of these studies were inconsistent. The majority of the studies combined colon cancer (CC) and rectal cancer (RC) samples as CRC for analysis. Furthermore, the small intestine, as the major site for immune surveillance in the gut, is understudied compared to the colon. Thus, the CRC heterogeneity puzzle is far from being solved, and more research is necessary for prospective trials that separately investigate CC and RC. Our prospective study aimed to map the colon cancer landscape using 16S rRNA amplicon sequencing in biopsy samples from the terminal ileum, healthy colon tissue, healthy rectal tissue and tumor tissue as well as in preoperative and postoperative stool samples of 41 patients. While fecal samples provide a good approximation of the average gut microbiome composition, mucosal biopsies allow for detecting subtle variations in local microbial communities. In particular, the small bowel microbiome has remained poorly characterized, mainly because of sampling difficulties. Our analysis revealed the following: (i) right- and left-sided colon cancers harbor distinct and diverse microbiomes, (ii) the tumor microbiome leads to a more consistent cancer-defined microbiome between locations and reveals a tumor microbiome–ileal microbiome association, (iii) the stool only partly reflects the microbiome landscape in patients with CC, and (iv) mechanical bowel preparation and perioperative antibiotics together with surgery result in major changes in the stool microbiome, characterized by a significant increase in the abundance of potentially pathogenic bacteria, such as *Enterococcus*. Collectively, our results provide new and valuable insights into the complex microbiome landscape in patients with colon cancer.

## 1. Introduction

Colorectal cancer (CRC) is the third leading cause of cancer death worldwide and the second leading cause of cancer mortality in Europe [1,2]. Although CRC incidence and mortality rates have decreased over the past decades, global trends have shown that the incidence among young adults aged 20–49 has increased [3]. While the prognosis of patients with early-stage disease is excellent, 40% of patients across all disease stages ultimately die from their disease within five years [4]. In the last decade, it became clear that differences in oncological outcome can be partly explained by differences in tumor biology. CRC is a highly heterogeneous group of tumors, and the pathogenesis of CRC is a complex and multifactorial process involving the accumulation of various genetic and epigenetic alterations [5]. Beyond these alterations, it is widely accepted that tumor location (right- and left-sided colon cancer, rectal cancer) is a crucial factor involved in disease progression as well as prognosis and influences disease management [6,7,8]. Right-sided colon cancer (RSCC) occurs within the cecum, ascending colon, hepatic flexure and transverse colon, while left-sided colon cancer (LSCC) arises in the splenic flexure, descending colon and sigmoid colon. The right- and left-sided colon have distinct embryological origins, developing from the mid- and hindgut [6]. Moreover, differences in clinical and molecular characteristics have been observed. In particular, RSCC presents with microsatellite instability, BRAF mutations, high immunogenicity and a worse prognosis [9,10]. To date, colon cancer (CC) and rectal cancer (RC) are synonymously termed CRC. However, based on experimental, translational and clinical research, there is more and more evidence to divide CC and RC as self-standing tumor entities [11].

Epidemiologic studies have identified a number of environmental factors that affect the risk of CRC carcinogenesis, including lifestyle, nutritional factors and the microbiome [12,13]. Microbes have been linked to cancer in 10–20% of cases [14,15,16]. Novel data have demonstrated that locations previously considered sterile, such as the liver, pancreas and even tumor tissue, harbor their own site-specific microbiome [17,18,19]. The human intestinal microbiome primarily comprises *Firmicutes, Bacteroidetes, Actinobacteria*, and *Proteobacteria* [16]. After rapid changes in the first years of life, the gut microbiome remains relatively stable for decades and displays gradual changes with advancing age. High diversity might be a key feature of a healthy microbiome [20,21,22,23,24]. Dysbiosis refers to an abnormality in the composition and/or function of the host’s symbiotic microbial ecosystem that exceeds its constitutive capacity and, as a result, has adverse effects on the host [25]. The microbiome can impact cancer initiation, progression and response to therapy [26,27,28]. In 2022, the microbiome is mentioned as a distinctive enabling characteristic for the acquisition of hallmarks of cancer capabilities [16,29]. The hallmarks of cancer, first published in 2000 by Hanahan and Weinberg and updated in 2011, are defined as a core set of functional capabilities acquired by human cells through their way to form malignant tumors [29,30]. Nevertheless, uncertainty remains regarding the direct and indirect effects of the microbiome in cancer [31,32,33]. Sequencing and association studies have demonstrated changes in microbial composition and ecology in patients with CRC, specifically, a decrease in commensal bacterial species (e.g., butyrate-producing bacteria) and the enrichment of opportunistic pathogens (e.g., proinflammatory). Moreover, there is strong evidence that the gut microbiome influences the efficacy of immune checkpoint inhibitors (ICIs) in CRC and other types of cancers [34,35,36,37,38,39]. However, recent studies have revealed that the ileal microbiota also determines the prognostic and predictive features and therapeutic responses of CC [40,41].

Altogether, most microbiome studies in CRC have focused on the analysis of fecal rather than tumor or mucosal samples. Furthermore, most studies have considered colon and rectal cancers as one disease entity, CRC. As mentioned before, obvious differences exist in tumor biology, molecular carcinogenesis, treatment and response to therapy. In surgical oncology, for decades researchers have been becoming aware that CC and RC are different diseases, based on multimodal treatments, surgical techniques, complication rates and relapse patterns. To overcome the heterogeneous nature of the microbiome, research should concentrate on separately describing results for CC and RC.

To address these aspects, we focused our attention on mapping associations between the microbiota and clinicopathologic features of tumor tissue and healthy tissue of the ileum, colon or rectum as well as fecal samples collected before and after surgery from patients with primarily untreated CC. The results provide a deeper understanding of the complex microbiome landscape in patients with colon cancer.

## 2. Results

### 2.1. Patient Characteristics

The study cohort consisted of 41 newly diagnosed, treatment-naive CC patients scheduled for elective surgery and included 23 male (56.1%) and 18 female (43.9%) patients, of whom 24 patients were with RSCC and 17 patients were with LSCC (only CC, no rectal cancer patients), aged from 39 to 90 years. The baseline clinical and pathological characteristics are shown in Table 1; no significant differences were observed between patients with RSCC and LSCC.

### 2.2. Microbiome Profile of the Study Cohort

#### 2.2.1. The Microbiome Landscape across the Locations

First, we analyzed all sample types (ileal tissue, healthy colon tissue, healthy rectal tissue, tumor tissue, preoperative stool and postoperative stool). The most abundant phyla in all samples were *Firmicutes* and *Bacteroidetes*, followed by *Actinobacteria*, *Proteobacteria, Verrucomicrobia* and *Fusobacteria*, to different degrees (Figure 1a). The microbiota profile at the genus level in all samples is shown in Figure 1b. The profile of the microbiota in the different analyzed samples differed from those found in the quality controls (mock community, water).

To estimate the richness and diversity of the different habitats, the alpha diversity indices were analyzed. We compared the Observed, Chao, ACE, Shannon, Simpson and Fisher indices of the different sample types at the genus level. The overall structure of the microbiota in the microhabitats was significantly different based on all indices: the Observed index (*p* value: 0.000002; (ANOVA) F value: 7.4813) (Figure 2a), the Chao1 index (*p* value: 0.00001; (ANOVA) F value: 6.4832), the ACE index (*p* value: 0.00004; (ANOVA) F value: 5.9738), the Shannon index (*p* value: 0.000000006; (ANOVA) F value: 10.664), the Simpson index (*p* value: 0.00000004; (ANOVA) F value: 9.6125) and the Fisher index (*p* value: 0.000002; (ANOVA) F value: 7.5182). The diversity was lowest in postoperative stool samples, which could be explained by the bowel preparation (mechanical and antibiotics) and surgical stress.

Moreover, a beta diversity analysis was performed. At the genus level, the analysis revealed that the overall structure of the microbiota in the analyzed habitats was significantly different (PCoA Jensen–Shannon (PERMANOVA) F value: 9.5743, R-squared: 0.22074, *p* value < 0.001; Figure 2b).

A linear discriminant analysis (LDA) coupled with effect size measurements (LEfSe) was applied to identify key taxa that were differentially abundant between the analyzed samples. A total of 46 key taxa were identified at the genus level (Figure 3, LDA score > 3, *p* value < 0.05, FDR-adjusted *p* value < 0.1; Appendix A).

#### 2.2.2. The Microbiome Communities Are Significantly Different between Tumor and Stool Samples

The early detection of CC is of great prognostic importance, and stool samples are a potential source of microbial biomarkers. We compared tumor tissue and preoperative stool samples and analyzed differences in the microbiota composition. The beta diversity comparisons showed significantly different bacterial community clusters between the tumor and stool samples (PCoA Jensen–Shannon divergence (PERMANOVA) F value: 18.721, R-squared: 0.19558, *p* value < 0.001, Figure 4a). The LEfSe analysis identified 35 genera whose abundances significantly differed between the tumor and stool samples (LDA score > 3, *p* value < 0.05, FDR-adjusted *p* value < 0.05; Figure 4b). No significant differences in the alpha diversity were observed between the tumor and stool samples (Figure 5a). The random forest classification machine learning algorithm was used to confirm the data. Using 120 trees, the algorithm achieved the best prediction with a classification error of 0.0253 (Appendix A). The top five ranked genera to discriminate between stools and tumors were Flavonifractor, Oscillibacter, Odoribacter, Roseburia and Eggerthella (Appendix A).

To determine whether the composition of the microbiome differs according to clinical factors, additional analyses were performed based on location (RSCC, LSCC) and pathologic parameters (T stage, differentiation, nodal stage, MSS status). The alpha diversity of the whole microbiome of the stool and tumor tissue was significantly different between the RSCC and LSCC groups (Observed index *p* value: 0.014561; (*t* test) statistics: 2.4996; Chao1 index *p* value: 0.017411; (*t* test) statistics: 2.4305). The MSS and MSI tumor groups were slightly but not significantly different (Chao1 index *p* value: 0.0508; (*t* test) statistics: −2.0505) (Figure 5b,c).

The tumor tissue of grade 3 tumors was significantly enriched in *Fusobacterium* and *Parvimonas*, while *Fusicatenibacter, Blautia, Intestimonas* and *Romboutsia* were significantly increased in grade 2 tumors (*p* value < 0.01, FDR-adjusted *p* value < 0.1). There was no significant difference among the T or N stages; we think that this was due to the stage-specific distribution: early T1/2 stages (*n* = 11) compared to T3/4 (*n* = 29) and more N-negative (*n* = 32) than N-positive patients (*n* = 9). In tumor tissue, no significant differences according to MSS status were observed.

In contrast, the preoperative stool of grade 2 patients was associated with *Dialister* and *Intestimonas*, while grade 3 tumors were significantly enriched in E. shigella (*p* value < 0.01, FDR-adjusted *p* value < 0.1). Furthermore, the stool of MSI patients was significantly enriched with *Clostridium_XIVb* (*p* value < 0.01, FDR-adjusted *p* value < 0.1). Taken together, these findings suggest that the stool microbiome (preoperative) only partly reflects the tumor microbiome.

The core microbiome, based on sample prevalence (>50%) and relative abundance (0.01%), is displayed in Figure 6. The core analysis revealed six genera as the core taxa across all samples. Among them, *Parabacteroides* was prevalent in more than half of the samples from the RSCC patients, while *Bifidobacterium* and *Roseburia* were prevalent in more than half of the LSCC patients. Taken together, these findings indicate that RSCC and LSCC harbored a diverse core microbiome, with *Bacteroides* as the predominant genus (Figure 6) in both.

#### 2.2.3. The Tumor Microbiome Profile: Significant Differences between RSCC and LSCC

For a deeper understanding of the intratumoral microbiome, we further analyzed the tumor tissue and sidedness (Figure 7). We first assessed the general tumor landscape. The top taxa in RSCC patients (Figure 7b) at the genus level were *Bacteroides* (15%), *Ruminococcus2* (10%), *Blautia* (8%), *Peptostreptococcus* (7%) and *Veillonella* (5%), and the top taxa in LSCC patients (Figure 7c) were *Blautia* (15%), *Bacteroides* (11%), *Streptococcus* (7%), *Parvimonas* (7%) and *Fusobacterium* (6%). The MSI patients (Figure 7d) harbored *Bacteroides* (18%), *Clostridium_XIVa* (11%), *Corprococcus* (9%) and *Blautia* (8%), while in the MSS patients (Figure 7e), the top taxa were *Bacteroides* (12%), *Blautia* (12%), *Ruminococcus2* (6%) and *Peptostreptococcus* (6%).

A comparison of alpha diversity revealed significant differences between RSCC and LSCC at the genus level. Based on the Chao1 (*p* value: 0.018981; (*t* test) statistics: 2.4735; Figure 8) and Observed (*p* value < 0.05) indices, the alpha diversity was significantly higher in LSCC than in RSCC (Figure 8a). There were no significant differences in alpha diversity based on sex, age, T stage, N stage or differentiation, while for the MSS status, these indices were significantly different (Chao1 index *p* value: 0.014618; (*t* test) statistics: −2.8349, Figure 8b).

The differential abundance analysis, which shows the highest power to compare groups, especially for less than 20 samples per group, revealed a significant increase in the abundance of *Haemophilus* and *Veilonella* in the tumor tissue of RSCC patients, while increased *Bifidobacterium, Akkermansia, Roseburia* and *Ruminococcus* were associated with LSCC (genus level, *p* value < 0.001, FDR-adjusted *p* value < 0.05). The FDR-adjusted LEfSe analysis revealed two significantly different genera, *Bifidobacterium* and *Romboutsia*, in LSCC patients (genus level, *p* value < 0.05, LDA > 3.0, FDR-adjusted *p* value < 0.05). The original LEfSe analysis revealed 10 significantly different genera: *Bifidobacterium, Romboutsia, Clostridium_III, Ruminococcus, Anaerostipes, Akkermansia, Clostridium_sensu_stricto* and *Asaccharobacter* in LSC patients and *Haemophilus* and *Veillonella* in RSCC patients (genus level, *p* value < 0.05, LDA > 3.0). In regard to MSS status, the original LEfSe analysis revealed seven significantly different genera: *Asaccharobacter, Actinomyces, Eubacterium, Pseudoflavonifractor, Fusicatenibacter* and *Anaerostipes* in tumor specimens from the MSS patients and Clostridium_III in tumor tissue from the MSI patients (genus level, *p* value < 0.05, LDA > 3.0). The FDR-adjusted LEfSe revealed no significant differences. The abundances of *Fusobacterium, Peptostreptococcus* and *Desulfotomaculum* were significantly different in grade 3 tumor specimens (original LEfSe, genus level, *p* value < 0.05, LDA > 3.0), but no significant differences were identified based on the FDR-adjusted *p* value (<0.05).

#### 2.2.4. The Microbiome of the Terminal Ileum: Tumor-Associated Alterations

We next assessed the general ileum landscape (Figure 9). The most abundant phylum was Firmicutes, followed by Bacteroidetes and Proteobacteria. The top 5 taxa at the family level were *Lachnospiraceae* (32%), *Streptococcaceae* (18%), *Bacteroidaceae* (8%), *Enterobacteriaceae* (8%) and *Verrucomicrobiaceae* (6%) (Figure 9a). The terminal ileum core microbiota, defined as genera with a threshold over 50%, are displayed in Figure 10. The typical ileal microbiota is dominated by the facultative anaerobic genus *Streptococcus* and the strict anaerobic genera *Bacteroides*, *Lachnospiraceae_incertae_sedis* and *Clostridium cluster XIV*.

From five LSCC patients, we also had specimens of the terminal ileum. In these patients, the top taxa at the genus level were *Streptococcus* (37% versus 10% in patients with RSC) and *Enterobacter* (19% versus 2% in patients with RSC). Due to the small sample size, no significant differences were observed in regard to sidedness. The core microbiome analysis further revealed that the ileal microbiome of the RSCC and LSCC patients as well as of the MSS and MSI patients harbored a diverse core microbiome (Figure 10c,d). The differential abundance analysis with the highest power to compare groups, especially for less than 20 samples per group, revealed five significantly different features for MSS status: *Enterobacter*, *Actinomyces* and *Streptococcus* for the MSS patients and *Eisenbergiella* and *Parasutterella* for the MSI patients. The original LEfSe analysis revealed three significantly different features, *Actinomyces, Abiotrophia* and *Atopobium*, in the MSS patients (*p* value < 0.05, LDA > 3.0), but the FDR-adjusted *p* values revealed no differences.

Between the ileal samples and preoperative stool samples, the alpha (Observed index *p* value < 0.01, [*t* test] statistics: −2.61) and beta diversity (PCoA Jensen–Shannon (PERMANOVA) F value: 18.525, R-squared: 0.23592, *p* value < 0.001) clustered significantly differently (Appendix A). The LEfSe analysis revealed 23 genera with a significantly different abundance (*p* value < 0.05, LDA > 3.0, FDR-adjusted *p* value < 0.05, Appendix A).

Next, we compared ileal samples and tumor tissue and interestingly did not reveal a significant difference in the alpha and beta diversity (Figure 11b). The original LEfSe analysis revealed only one significantly different abundant genus, *Atopobium* (*p* value < 0.05, LDA > 3.0), in specimens of the terminal ileum, and the FRD-adjusted analysis (<0.05) revealed no significant differences.

Additionally, between the ileal samples and healthy colon tissue samples, no significant differences in alpha diversity were observed, while the beta diversity was significantly different (PCoA Jensen–Shannon (PERMANOVA) F value: 3.8652, R-squared: 0.063505, *p* value < 0.03; [PERMDISP] F value 4.7804, *p* value < 0.03; Figure 11a). The LEfSe analysis revealed three genera with significantly different abundances in specimens of the terminal ileum: *Streptococcus*, *Gemella* and *Granulicatella* (*p* value < 0.05, LDA > 3.0).

#### 2.2.5. The Stool Microbiome Structure: Sequential Analysis before and after Surgery Revealed Major Changes

Due to bowel preparation, perioperative antibiotic prophylaxis and surgery, the stool microbiome underwent major changes before and after surgery. The ratio between *Firmicutes* and *Bacteroidetes* (regarded as dysbiosis) was decreased: the preoperative stool samples harbored 41% *Firmicutes* and 51% *Bacteroides*, while the postoperative samples consisted of 29% *Firmicutes* and 60% *Bacteroides* (Figure 12a). The microbiome composition differed strikingly at the genus level between the timepoints (beta diversity analysis (PERMANOVA) F value: 14.506; R-squared: 0.18019; *p* value < 0.001) (Figure 12c). Bacterial richness and evenness were significantly lower in the postoperative stool samples, and the postoperative stool samples were characterized by a significant increase in the abundance of *Enterococcus* (*p* value < 2.20 × 10^9^), LDA –5.84), a lactic-acid-producing bacterial genus that includes potentially pathogenic strains.

## 3. Discussion

To date, CC and RC are regarded as a single disease entity, termed CRC. Biologically, CRC is a heterogeneous group of tumors, characterized by high interpatient and intratumor heterogeneity with variable clinical features and outcomes. Tumor sidedness (CC versus RC) is one aspect of heterogeneity, and it correlates with distinct biological and molecular characteristics, as well as with different disease management strategies. In surgical clinical oncology, for decades researchers have been becoming aware of the fact that CC and RC are different diseases, based on their surgical procedures and challenges, as well as complication rates and local recurrence patterns.

We believe that understanding CRC heterogeneity and regarding CC and RC as two different tumor entities is fundamental to overcoming the inconsistent study results and the heterogeneous nature of microbiomes. Thus, our prospective, observational study aimed to characterize the microbiome landscape of different body sites in patients with treatment-naive CC. Moreover, we correlated the microbiome with sidedness (RSCC versus LSCC) and other clinicopathologic features of tumor progression (such as stage, lymph node involvement and tumor grade). Right or extended right hemicolectomy with complete mesocolic excision involves the resection of the tumor along with nonmalignant tissues, including the terminal ileum. These procedures provide surgical access to the ileal lumen. Studies investigating the bacterial composition of CC via a comparison of matched samples from multiple locations in the body, such as feces, tumor tissue and normal-healthy mucosa tissue, are rare and have reported inconsistent results. As mentioned before, one reason might be that the majority of these studies combined CC and RC samples as CRC for their analyses. Furthermore, analyzing the gut microbiome using stools does not capture all the microbes in the gut, in particular mucosally adherent microbes and microbes in the small intestine (ileal microbiota). Most of our knowledge has been derived from studies of ileal biopsies during colonoscopies or naso-ileal catheters. However, data must be interpreted cautiously because accessing the ileal microbiome via retrograde examinations is prone to contamination [34,35,40,42,43,44,45]. The ileal microbiota is oral-like and more variable than its colonic counterpart, and across several studies, the ileal core microbiome is constituted by *Firmicutes*, *Proteobacteria* and *Actinobacteria*. Villmones et al. reported the ileal microbiome of 27 patients based on samples collected during radical cystectomies with urinary diversion. They demonstrated that the distal part of the ileum harbors a distinct niche that differs from the colonic flora. The REIMAGINE study revealed that the stool microbiome was a good proxy for that of the large intestine but differed substantially from that of the small intestine [19]. The Zitvogel and Roberti group recently reported that the ileal immune tonus was affected by colonic carcinogenesis in RSCC, indicated by the fact that the growth of heterotopic or orthotopic CCs induced this upregulation of ileal immune gene products [34,35]. They also demonstrated that the ileal microbiome governed the efficacy of chemotherapy and PD-1 blockade in CC independent of microsatellite instability. Our study further reveals that the presence of a colonic tumor leads to a more consistent cancer-defined microbiome and shapes the normal spatial heterogeneity existing along the intestinal tract. No significant differences in alpha or beta diversity were identified between the ileal samples and tumor samples. In our cohort, due to operational reasons, in several cases of LSCC, an extended operation was needed, and from those patients, we also collected terminal ileum specimens. Interestingly, also in this subgroup, we did not observe a significant difference in beta diversity between the tumor and ileum. *Bifidobacterium* was significantly associated with LSCC and was found in the core microbiome of more than half of the ileal samples of the LSCC patients. In contrast, *Bifidobacterium* was not found to be a core microbiota in the ileal samples of the RSCC patients. Furthermore, the abundance profile of the terminal ileum revealed that samples from the patients with LSCC harbored 37% *Streptococcus* and 19% *Enterobacter*, while samples from the patients with RSCC harbored 10% *Streptococcus* and only 2% *Enterobacter*. The subgroup of the MSI patients harbored 20% *Akkermansia* and 25% *Streptococcus*, while in the MSS patients, the percentage of *Akkermansia* was less than 1%, and the amount of *Streptococcus* was 8%. A limitation of the study is that the microbiome shifts might have been induced by the preoperative bowel preparation, but colonoscopy sampling also requires bowel preparation.

However, we know that the fecal microbiota differs from the microbiota of mucosal tissue in regards to oxygen and nutrition needs [46]. Analyzing the bacterial composition, especially the similarity or dissimilarity, between tumors, healthy mucosa and stool from the same individual provides information regarding changes in the microenvironment that have occurred that favor growth in the right- or left-sided colon. Most microbiota identified from human feces belong to the phyla *Firmicutes, Bacteroidetes, Proteobacteria, Actinobacteria* and *Verrucomicrobia*, with 90% belonging to *Firmicutes* or *Bacteroidetes*. Although a disease-specific microbiota signature has yet to be identified, patients with CRC have reduced bacterial diversity and richness compared to healthy individuals. Specific bacteria, such as Fusobacterium nucleatum, as well as certain *Bacteroides fragilis* and *E. coli* species, are known CRC-associated pathobionts. We confirm previous observations that CC tumors harbor orally derived opportunistic pathogens [47,48]. Furthermore, we observed that the stool microbiome only partially reflects the tumor microbiome. We identified 35 genera whose abundance significantly differed between tumor and stool samples. Interestingly, between paired tumor and nontumor healthy colon tissue, no significant differences were observed. We think the presence of a colonic tumor leads to a more consistent microbiota profile. This finding supports previous studies by Murphy et al. and Liu et al., which demonstrated that the microbiotas in tumor tissue and normal mucosa tissue of patients with CRC were similar [49,50]. We further observed that RSCC and LSCC patients harbor distinct microbiomes, characterized by differences in microbial diversity and bacterial taxa. The alpha diversity in the LSCC patients was significantly higher than that in the RSCC patients. Consistent with our results, Phipps et al. showed that patients with RSCC showed fewer taxonomic differences than those with left-sided carcinomas [51]. However, unlike our study, the study of Phipps et al. included rectal cancer patients. Furthermore, we were able to show that the tumor tissue of RSCC patients was characterized by a significant increase in the abundances of *Haemophilus* and *Veilonella*, while increased abundances of *Bifidobacterium* and *Ruminococcus* were associated with LSCC. Overall, grade 3 tumors were significantly enriched in *Fusobacterium* and *Parvimonas*. Little is known about *Parvimonas,* but interestingly, *Parvimonas micra* and *Fusobacterium* have been shown to aggregate and form biofilms in vitro [52,53]. Biofilm formation is linked to inflammatory bowel disease and CC. Due to dysbiosis, biofilm formation occurs within the inner mucus layer, normally free from microorganisms, which could result in direct contact between bacteria and epithelial cells [54]. As mentioned in the introduction, CRC numbers are rising in younger people worldwide. The increased incidence of early-onset CRC can be the consequence of environmental influences (e.g., having a Western diet, food quality and additive-laden food). Early onset is more frequent in left-sided colon. We consider the microbiome of someone developing colorectal cancer at an age over 80 years to be different from someone with early-onset colorectal cancer. Unfortunately, we have too few patients in the “younger age” group for a detailed analysis. In line with the group of P. O’Toole, we recommend adjusting for age to improve the identification of gut microbiome alterations in multiple diseases.

Accumulating evidence suggests a critical role of intestinal dysbiosis in surgical site infections and anastomotic leakage after CRC surgeries. Despite improvements in surgical techniques, new energy devices and intensive care management, anastomotic leakage is still a significant problem in daily clinical practice. We recently linked the microbiome to surgical complications in pancreatic surgeries [18]. In CRC surgeries, the microbiome has also been linked to postoperative complications [55,56,57]. Many factors beyond geography, diet and lifestyle affect tumors, independent of the microbiome composition, prior gastrointestinal surgery, antibiotic treatment or preoperative bowel preparation regimen. To prevent this type of possible bias, we designed a study in which all patients received the same preoperative bowel preparation regimen on the day prior to their surgery and perioperative antibiotic prophylaxis (most of the studies did not even report this treatment). Furthermore, we excluded upfront confounding variables, such as antibiotic usage, four weeks prior to surgery and systemic conditions related to bowel dysfunction, and only patients who lived in Franconia (Germany) for at least six months were included. We observed that the pre- and postoperative stool microbiomes differed strikingly. Bacterial richness and evenness were significantly lower in the postoperative stool samples. Furthermore, postoperative stool samples were highly dysbiotic and characterized by a significant increase in the abundance of *Enterococcus*, a potentially pathogenic bacterium. These findings suggest that bowel preparation, perioperative antibiotic treatment and surgery had a major effect on the stool microbiome. We aim to begin a new study analyzing the impact of mechanical bowel preparation on the intestinal microbiome in the context of surgery and outcomes.

## 4. Materials and Methods

### 4.1. Study Design

This study population consisted of 41 patients from the prospective Erlanger microbiome study, an observational trial approved by the local ethics committee (Protocol Number: 420_18 B). Treatment-naive patients undergoing elective surgery for histologically proven or suspected CC were screened for eligibility for study participation. Patients with antibiotic therapy within 4 weeks prior to surgery, diseases significantly affecting gastrointestinal function (Crohn’s, Ileus) and patients who needed emergency surgery were excluded. Each patient received the same mechanical oral bowel preparation and a standardized single shot of a 3rd generation cephalosporine and metronidazole approximately 30 min before the surgical procedure. The participants were prospectively recruited between 2018 and 2019. CC tumor samples and paired healthy mucosal tissue samples of the proximal resection margin (terminal ileum or healthy colon) and distal resection margin (healthy colon or healthy rectal tissue) of the resected specimen were obtained intraoperatively. Preoperative and postoperative stool samples were self-collected by the patients according to a well-explained protocol.

### 4.2. Sample Processing and DNA Purification

Stool samples were collected and stabilized before surgery and bowel preparation (stool preOp) and after surgery (stool postOp) on days 5–7 using the Omnigene Gut system (DNA Genotek, Ottawa, ON, Canada) and stored at −80 °C until DNA extraction. DNA was extracted from stool using the PSP Stool DNA stool kit according to the specifications of the manufacturer (Invitek Molecular, Berlin, Germany). Specimens of tumor tissue and mucosal tissue were collected immediately after resection, suspended in Qiagen RNA later buffer and stored at −80 °C. DNA from tumor tissue and mucosal tissue of the proximal and distal resection margins was extracted using Dulbecco’s phosphate buffered saline (Sigma Aldrich Chemistry GmbH, St. Louis, MO, USA) and the Qiamp Microbiome Kit (Qiagen, Hilden Germany) according to the manufacturer’s recommendations. DNA from stool samples was extracted using a PSP^®^ Spin Stool DNA Kit (Invitek Molecular) and LookOut^®^ DNA Erase (Sigma Life Science, St. Louis, MO, USA). DNA was subsequently quantified using a Qubit device (Thermo Fisher Scientific, Waltham, MA, USA).

### 4.3. 16S rDNA Amplification

The V3+4 region of the 16S rRNA gene was amplified using 10 ng of bacterial template DNA with degenerate region-specific primers (341F: 5′-ACTCCTACGGGAGGCAGCAG-3′ and 806R: 5′-123 GGACTACHVGGGTWTCTAAT-3′), containing barcodes and Illumina flow cell adaptor sequences [58], in a reaction consisting of 25 (stool) or 35 (tissue) PCR cycles (98 °C 15 s, 58 °C 20 s, and 72 °C 40 s) using the NEBNext Ultra II Q5 Master Mix (New England Biolabs, Ipswich, MA, USA). Amplicons were purified with Agencourt AMPure XP Beads (Beckmann Coulter, Brea, CA, USA), normalized and pooled before sequencing on an Illumina MiSeq device using a 600-cycle paired-end kit and the standard Illumina HP10 and HP11 sequencing primers.

### 4.4. Bioinformatic Processing of the Sequencing Data

For bioinformatic processing, the terminal 15 bases of both forward and reverse reads were removed before merging and quality filtering using fastq_mergepairs and fastq_filter_options from Usearch 10 [58]. Subsequently, merged fastq files were demultiplexed and trimmed using Cutadapt [59]. For 16S sequence determination, the Uparse and Sintax algorithms within Usearch using the Silva 16S rRNA database (v123) were applied. All reads were mapped to OTUs, and an OTU table was created using a Qubit device (Thermo Fisher Scientific). The V3+4 region of the 16S rRNA gene was amplified using 10 ng of bacterial template DNA with degenerate region-specific primers (341F: 5′-ACTCCTACGGGAGGCAGCAG-3′; 806R: 5′-123 GGACTACHVGGGTWTCTAAT-3′) containing barcodes and Illumina flow cell adaptor sequences in a reaction consisting of 25 (stool) or 35 (tissue) PCR cycles (98 °C 15 s, 58 °C 20 s, 72 °C 40 s) using the NEBNext Ultra II Q5 Master Mix (New England Biolabs, Ipswich, MA, USA). Amplicons were purified with Agencourt AMPure XP Beads (Beckmann Coulter, Brea, CA, USA), normalized and pooled before sequencing on an Illumina MiSeq device using a 600-cycle paired-end kit and the standard Illumina HP10 and HP11 sequencing primers. For bioinformatic processing, the terminal 15 bases of both forward and reverse reads were removed before merging and quality filtering using fastq_mergepairs and fastq_filter_options from Usearch 10 [58]. Subsequently, merged fastq files were demultiplexed and trimmed using Cutadapt [59]. The 16S Uparse and Sintax [60] algorithms were performed within Usearch using the silva 16S rRNA database (v123) [61,62].

### 4.5. Microbiome Analyses

The Microbiome Analyst platform [63,64] was used to calculate alpha and beta diversities and to compare the relative abundance of bacterial taxa. For richness measurements, we used Observed (amount of unique OTUs found in each sample) and Chao1 (also accounting for unobserved species based on low-abundance OTUs). For evenness measurement, Shannon diversity was used, which accounts for both richness and abundance. A *p* value of <0.05 was considered significant. Beta diversity represents the diversity between microbial communities. Bray–Curtis dissimilarity or the Jensen–Shannon distance was calculated to measure beta diversity, and then, principal coordinates analysis (PCoA) was applied for visualization. For differential analysis, DESeq2 was used for samples less than 20 samples per sample; it is computationally intensive but more robust with low false positive rates, and a *p* value of <0.05 was considered significant.

Linear discriminant analysis effect size (LEfSe) was used to identify the key microbial taxa associated with the different locations. This analysis integrates statistical significance with biological consistency (effect size) estimation. It uses a nonparametric factorial Kruskal–Wallis (KW) rank sum test to detect features with significant differential abundance with respect to the class of interest, followed by linear discriminant analysis to estimate the effect size of each differentially abundant feature. The original LEfSe implementation uses original *p* values when determining significant taxa, and an LDA score > 3 (effect size) and a *p* value of <0.05 were considered statistically significant. Meanwhile, the Microbiome Analyst implementation provides the option to use either original or FDR-adjusted *p* value cutoffs to identify significant features.

## 5. Conclusions

In summary, our findings provide the following new insights. RSCC and LSCC harbor distinct niches and have different microbiome compositions. The presence of a colonic tumor leads to a more consistent cancer-defined microbiome and shapes the normal spatial heterogeneity existing along the intestinal tract. The tumor microbiome may contribute towards shaping a favorable microbiome across the large intestine border into the ileum and also in LSCC. The stool microbiome only partly reflects the microbiome landscape of patients with CC. Mechanical bowel preparation and perioperative antibiotics together with surgery resulted in major changes in the stool microbiome, characterized by a significant increase in the abundance of potentially pathogenic bacteria, such as *Enterococcus*.

We believe that regarding CC and RC as two different tumor entities is fundamental to overcoming the inconsistent study results and the heterogeneous nature of microbiomes. Overall, our results have implications for understanding the role and impact of the microbiome in right- and left-sided CC.

## Figures and Tables

**Figure 1 ijms-24-03265-f001:**
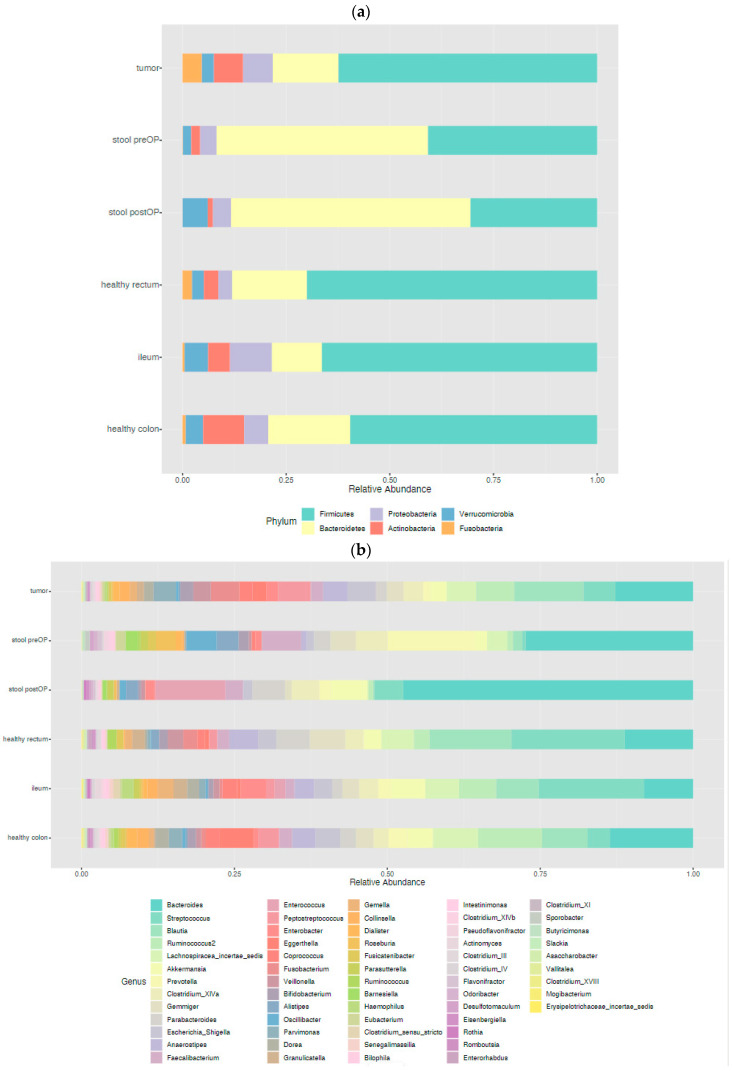
Taxonomic analysis of the microbiome in the different habitats of CC patients: represented at the phylum level (**a**) and genus level (**b**).

**Figure 2 ijms-24-03265-f002:**
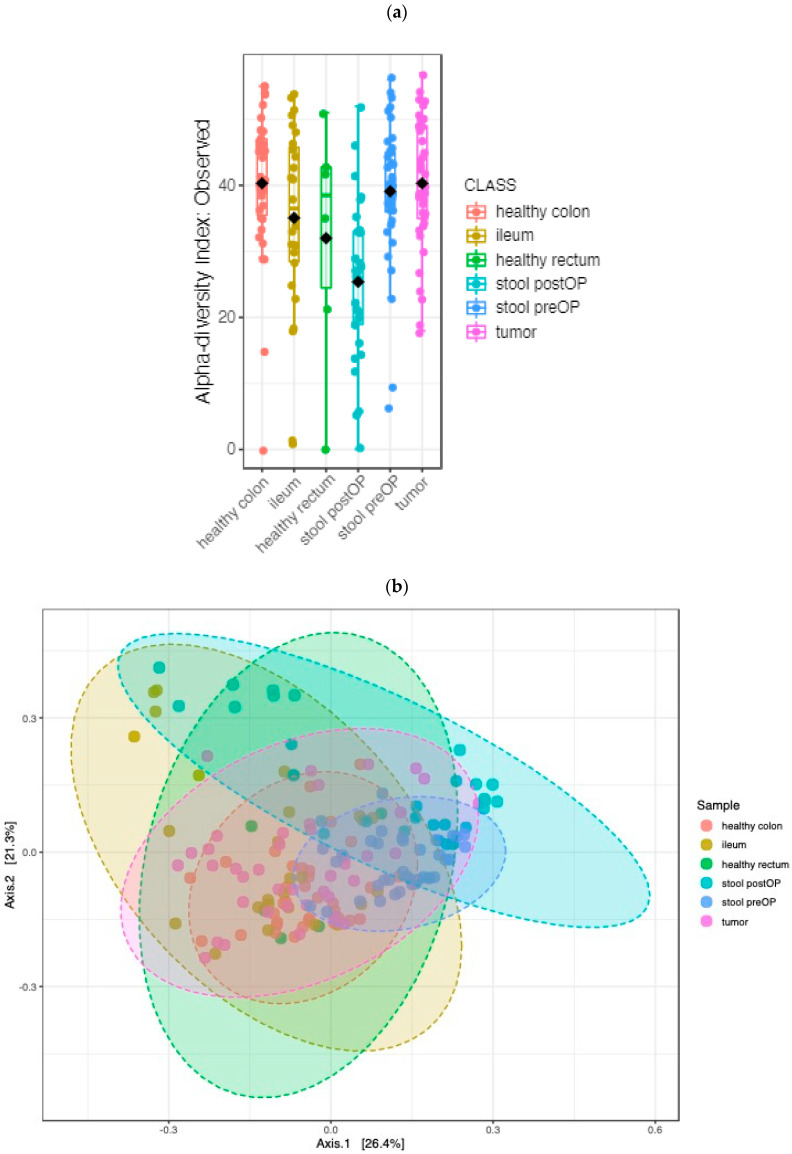
Microbiome diversity comparison between the locations of CC patients: alpha diversity box plot (Observed, *p* value < 0.001) (**a**) and principal coordinate analysis (PCoA) using Jensen–Shannon metric distances of beta diversity (**b**) at the genus level, *p* value < 0.001.

**Figure 3 ijms-24-03265-f003:**
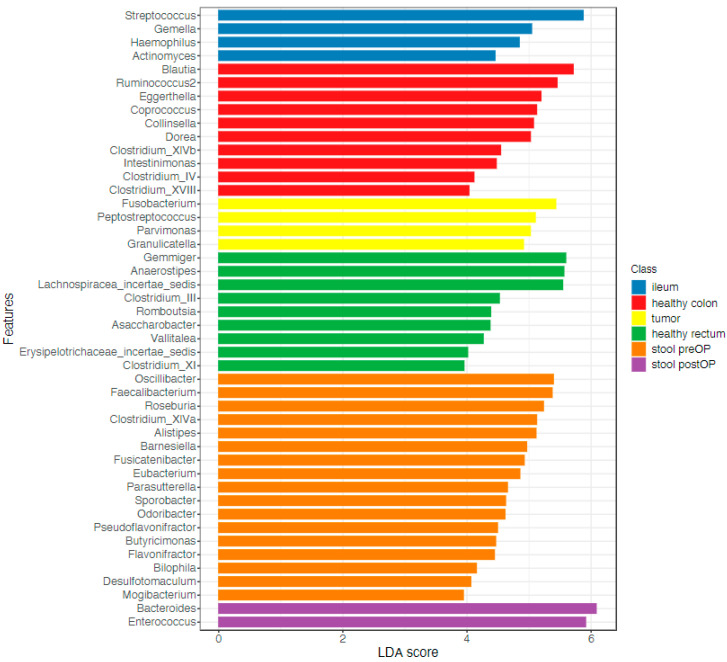
Microbiome communities are significantly different between the locations: LEfSe analysis computed from genera differentially abundant in the analyzed microhabitats, *p* value < 0.05.

**Figure 4 ijms-24-03265-f004:**
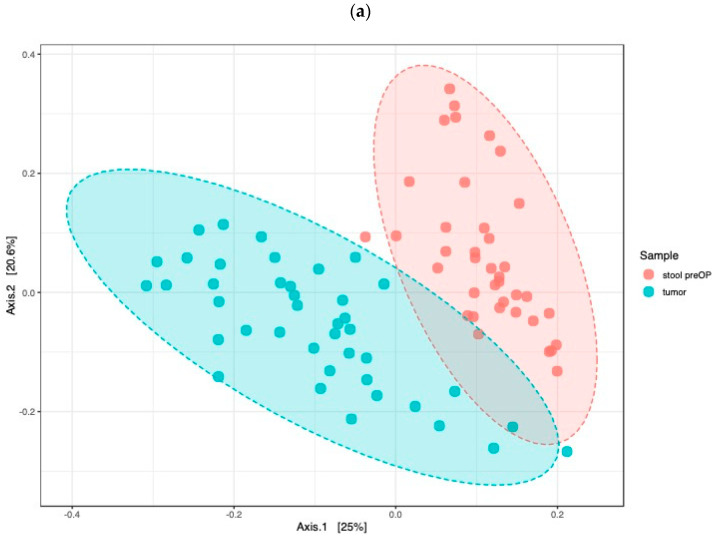
Stool microbiome only partly reflects the microbiome landscape in CC patients: PCoA using Jensen–Shannon divergence of beta diversity between tumor and preoperative stool, *p* value < 0.001 (**a**), LEfSe detected marked differences in the predominance of bacterial communities between tumor and preoperative stool, *p* value < 0.05 (**b**).

**Figure 5 ijms-24-03265-f005:**
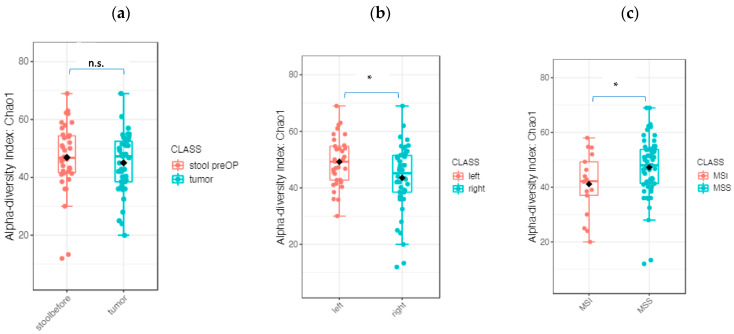
Microbiome composition according to tumor sidedness and MSS status: (**a**) diversity analysis using the Chao1 alpha diversity index between tumor and preoperative stool; (**b**) overall microbiome of the tumor and preoperative stool according to sidedness RSCC and LSCC; and (**c**) overall microbiome of the tumor and preoperative stool according to MSS status (* *p* < 0.05, n.s, not significant).

**Figure 6 ijms-24-03265-f006:**
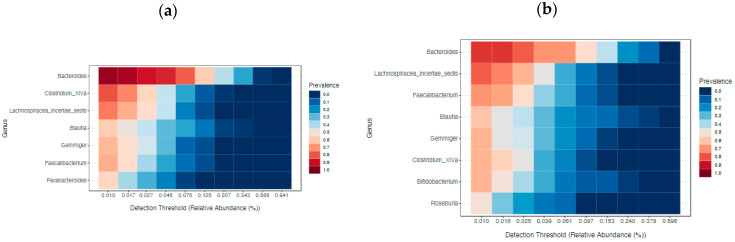
Heatmap of the core microbiome: the overall core microbiome of stool and tumor between RSCC (**a**) and LSCC patients clustered differentially (**b**).

**Figure 7 ijms-24-03265-f007:**
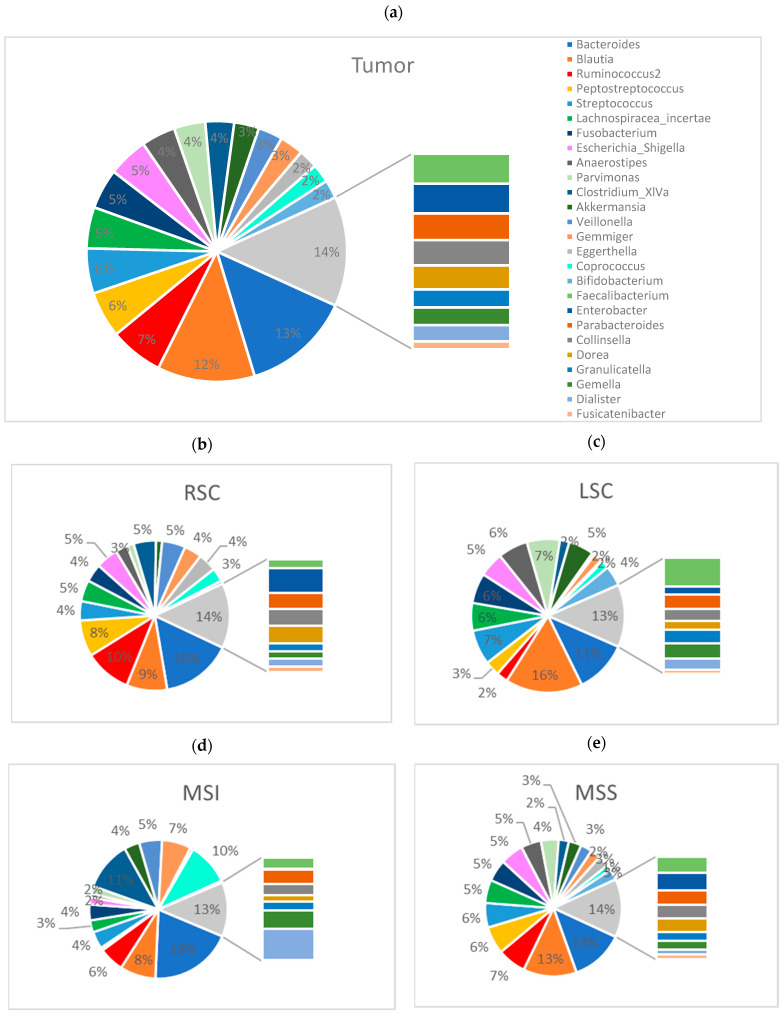
Taxonomic analysis of the tumor microbiome composition: Pie chart showing the abundance profile of the tumor samples (**a**), RSCC subgroup (**b**), LSCC subgroup (**c**), MSI subgroup (**d**) and MSS subgroup (**e**) at the genus level.

**Figure 8 ijms-24-03265-f008:**
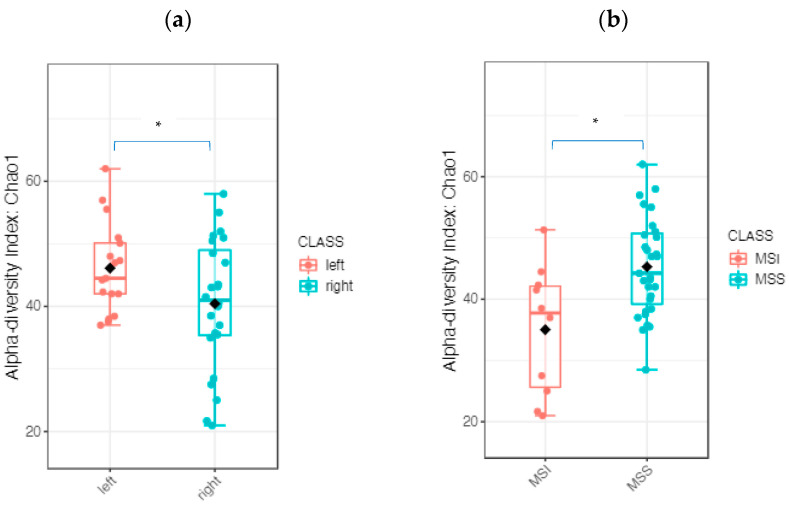
Tumor microbiome diversity comparison: the alpha diversity analysis revealed significant differences between RSCC and LSCC (**a**) and between MSS and MSI patients (**b**) at the genus level (* *p* < 0.05).

**Figure 9 ijms-24-03265-f009:**
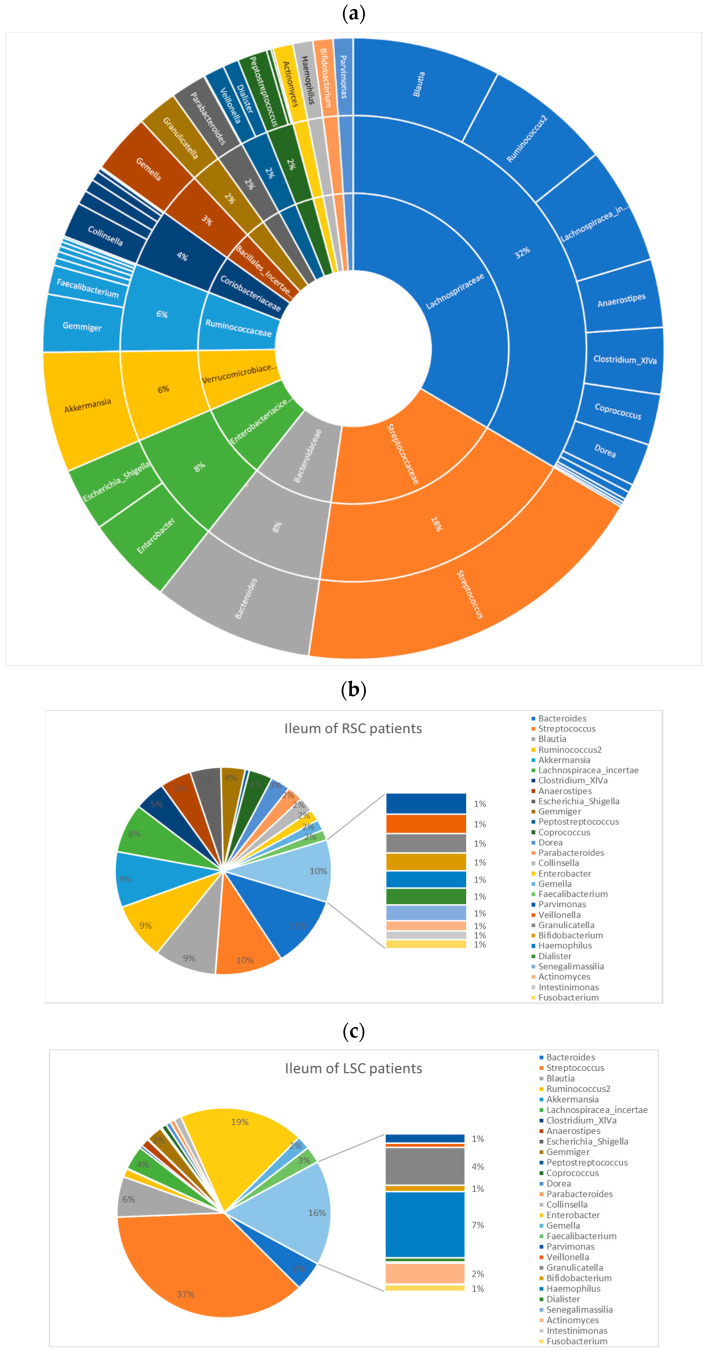
The ileal microbiome profile, taxonomic analysis: (**a**) Pie chart of the microbiome abundance profile of the terminal ileum. The inner circle represents the family level, and the outer circle represents the genus level. Microbiome profiles classified according to tumor location and MSS status: RSCC (**b**), LSCC (**c**), MSS (**d**) and MSI (**e**).

**Figure 10 ijms-24-03265-f010:**
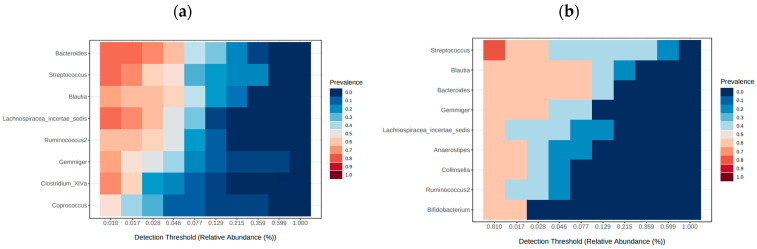
Heatmap of the core microbiome of the terminal ileum (defined as genera present in >50% of samples), based on tumor location: RSCC (**a**) and LSCC (**b**), and on MSS status: MSI (**c**) and MSS (**d**).

**Figure 11 ijms-24-03265-f011:**
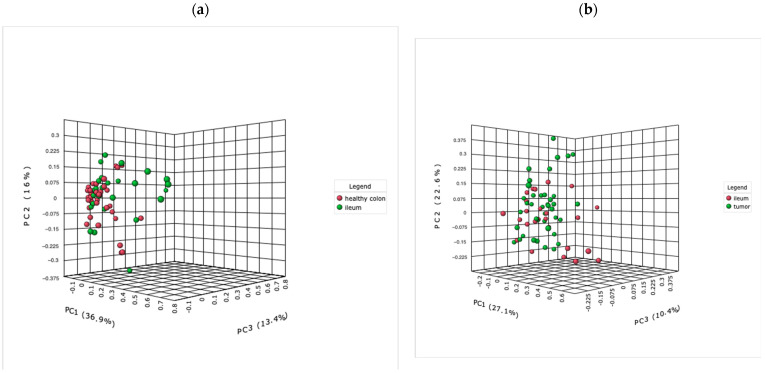
The tumor microbiome–ileal microbiome association: PCoA using Jensen–Shannon divergence of beta diversity between ileal and healthy colon tissue was significantly different, *p* value < 0.05 (**a**), while no significant differences were observed between ileal samples and tumor samples (**b**).

**Figure 12 ijms-24-03265-f012:**
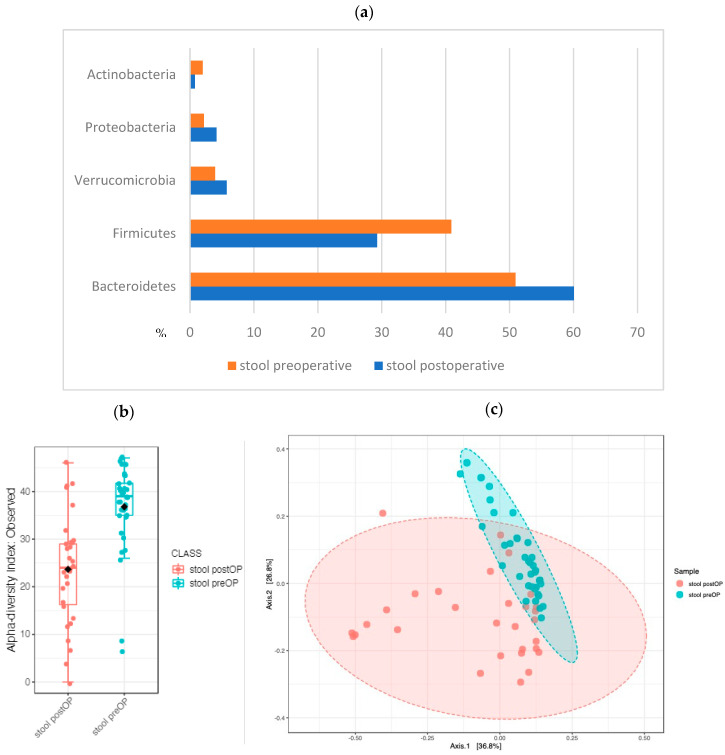
The stool microbiome preoperative and postoperative showed major differences: (**a**) Phylum level abundance profile of preoperative and postoperative samples. Comparison of pre- and postoperative stool revealed significant differences at the genus level: (**b**) alpha diversity and (**c**) beta diversity, (*p* < 0.001).

**Table 1 ijms-24-03265-t001:** Clinicopathological characteristics of the study participants.

Patient Characteristics	Right-Sided Colon Cancer n (%)	Left-Sided Colon Cancer n (%)	*p* Value
Age ○ overall, years ○ mean, years ○ group 1: ≤60 years ○ group 2: 61–79 years ○ group 3: ≥80 years	40–90 70.9 3 (12.5) 16 (66.7) 5 (20.8)	39–88 67.4 5 (29.4) 10 (58.8) 2 (11.8)	0.36
Sex ○ Male ○ Female	12 (50) 12 (50)	11 (64.7) 6 (35.3)	0.35
BMI ○ <18.5 Underweight ○ 18.5–24.9 Normal weight ○ 25.0–29.9 Preobesity ○ 30.0–34.9 Obesity class 1 ○ 35.0–39.9 Obesity class 2 ○ ≥40 Obesity class 3	0 11 (45.8) 9 (57.5) 3 (12.5) 1 (4.2) 0	0 4 (23.5) 8 (47.1) 4 (23.5) 1 (5.9) 0	0.50
Dietary patterns ○ omnivorous ○ vegan/vegetarian	23 (95.8) 1 (4.2)	16 (94.1) 1 (5.9)	0.80
Smoking ○ yes ○ no	2 (8.3) 22 (91.6)	3 (17.6) 14 (82.4)	0.37
Medication ○ no ○ yes (1–3) ○ yes (3–5) ○ yes (>5)	3 (12.5) 12 (50) 4 (16.7) 5 (20.8)	4 (23.5) 6 (35.3) 4 (23.5) 3 (17.6)	0.68
T-stage ○ 1 ○ 2 ○ 3 ○ 4	3 (12.5) 6 (25) 11 (45.8) 4 (16.7)	2 (11.8) 0 14 (82.4) 1 (5.9)	0.06
N-stage ○ 0 ○ 1 ○ 2	19 (79.2) 4 (16.7) 1 (4.2)	13 (76.5) 4 (23.5) 0	0.62
Differentiation (G) ○ 1 (well differentiated) ○ 2 (moderately differentiated) ○ 3 (poorly differentiated)	0 11 (45.8) 13 (54.2)	0 9 (52.9) 8 (47.1)	0.65
R-status ○ local R0	24	17	n/a
M-status ○ 0 ○ 1	21 (87.5) 3 (12.5)	15 (88.2) 2 (11.8)	0.94
MSS status ○ MSS ○ MSI	16 (66.7) 8 (33.3)	15 (88.2) 2 (11.8)	0.11

## Data Availability

The data are available on reasonable request from the corresponding author.

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
