# Peer review of "Colon Cancer Microbiome Landscaping: Differences in Right- and Left-Sided Colon Cancer and a Tumor Microbiome-Ileal Microbiome Association"

_ijms, 2023, doi:10.3390/ijms24043265_

Round 1

Reviewer 1 Report

the Authors present the results of a very good study exploring the role of microbiota landscape in CRC. The rationale is very deep and well presented, and the results may have hypothesis-generating relevance.

I think that this study does deserve dissemination to the public domain. Just a couple of minor suggestions:

- I understand that this kind of study cannot be conducted on a large sample of patients. However, can the Authors better clarify the hypothesis-generating nature of this study and the potential relevance to clinical practice? this would help, in my opinion, give even more relevance to the study findings.

- the Graphical abstract is unclear to me, can the Authors see if it deserves improvement?

Author Response

We thank the reviewer for the suggestions. The manuscript was send to an english editing service, we uploaded the certificate.

We added some points in order to clarify our purpose: As mentioned, our prospective study looked at colon cancer from several sides (stool, tumor samples versus healthy small bowel and large bowel tissue, right versus left, preoperative versus postoperative) and aims to develop a deeper understanding of colon cancer, in particular in combination with detailed host factors. Furthermore, we believe that for understanding CRC heterogeneity it is important to regard CC and RC as two different tumor entities (p3, 98ff).  We are presenting one of the first studies dealing with a detailed analysis of different samples, including the terminal ileum.  It is necessary to understand the different aspects of the microbiome and the contribution to cancer development. Furthermore, our study analyzes postoperative changes in the stool microbiome. However, stool does not capture all the microbes in the gut, in particular microbes in the small intestine and mucosally adherent microbes (added p20, 590ff). Thus, we think stool samples are one aspect of the microbiome puzzle and tumor as well as healthy tissue samples a long with detailed patient information are another important aspect (p 21 630ff).  

The graphical abstract shows the general workflow of our study: patients, sample collection (stool, mucosal samples), sequencing and results. We have separately uploaded high resolution versions of all figures including the graphical abstract.

Reviewer 2 Report

Dr Kneis and team present an observational study on patients with CRC and describe the microbiome in these patients.  The study is relevant as it may present some mechanistic insights into disease-heterogeneity as to location and biological behaviour. 

The introduction provides sufficient information. The results are presented clearly and the conclusion are backed by data presented. 

The discussion should be shortened somewhat. A brief discussion of mechanistic and cellular pathways of differing microbiota in CRC progression could be helpful.

Author Response

We thank the reviewer for the comments. The manuscript was send to an english editing service and we have uploaded the certificate. The discussion was revised and shortened in several paragraphs. Since the other reviewer wanted some aspects described in more detail, these suggestions had also to be incorporated despite the shortening of the discussion.

Reviewer 3 Report

This paper studies the difference in microbiome compositions in right-sided colon cancer (RSCC) and left-sided colon cancer (LSCC). Amplicon sequencing of the 16S rRNA gene was performed on biopsy samples from the terminal ileum, healthy colon tissue, healthy rectal tissue, tumor tissue, and stool samples of patients. The results showed RSCC and LSCC exhibit distinct microenvironmental niches and possess distinct microbiome compositions. This is an interesting study. I have several comments.

1. You divided patients into three groups ( group 1: ≤ 60 years, group 2: 61-79 years, and group 3: ≥ 80 years). It is not clear why you divided the patients into these three groups. Since the mean ages are around 70, why do you not consider two groups: group 1: ≤ 69 years, group 2:  ≥ 70 years? If you only consider these two groups, can you still have a significant result?

2. In Supplementary file S2, are these top genera used to differentiate fecal and tumor samples, the dominant genera for the human intestinal microbiome? Do these top genera play an important role in initiating colon cancer? The stool sample test is usually used as a screening tool for colon cancer. Do your results indicate that the stool sample test is still an effective screening tool?

3. Lines 97 and 98. You mentioned, “most microbiome studies in CRC have focused on the analysis of fecal rather than tumor or mucosal samples”.  It is easier to obtain fecal samples than tumor or mucosal samples, and it is more practical to test fecal samples than mucosal samples. From this viewpoint, studying fecal samples may be more useful than tumor or mucosal samples. It is not clear how this study can be applied to diagnose or prevent colon cancer.

4. The quality of Figure 2 needs improvement. In addition, the notations (a) and (b) in Figure 2 should be put on top of the figures.

Author Response

We thank the reviewer for the helpful comments. Please find attached our reply. The manuscript was send to an english editing service and we have uploaded the certificate. 

  1. You divided patients into three groups (group 1: ≤ 60 years, group 2: 61-79 years, and group 3: ≥ 80 years). It is not clear why you divided the patients into these three groups. Since the mean ages are around 70, why do you not consider two groups: group 1: ≤ 69 years, group 2:  ≥ 70 years? If you only consider these two groups, can you still have a significant result?

Colorectal cancer numbers are rising in younger people worldwide. The change in the incidence of disease within generations suggests the reason to lie in environmental influences.

Age was not our primary focus, but we know from other studies of our department that colorectal cancer in younger patients and older patients is different (DOI: 10.1007/s00384-011-1291-8; DOI: 10.1055/s-0032-1328570; 10.1055/a-0591-6283).

The group of P. O´Toole recently recommended to adjust for age to improve the identification of gut microbiome alterations in multiple diseases (DOI: 10.7554/eLife.50240). Thus, we divided our cohort into three different age groups. We consider the microbiome of someone developing colorectal cancer at an age over 80 years to be different from someone with early-onset colorectal cancer. Unfortunately, we have too few patients in the “younger age” group. This will be part of a future study, however, given the evidence, we think it is important to report and differentiate according to the various, admittedly arbitrary, age groups. We have added this aspect on p 21 line 663ff of the manuscript.

  1. In Supplementary file S2, are these top genera used to differentiate fecal and tumor samples, the dominant genera for the human intestinal microbiome? Do these top genera play an important role in initiating colon cancer? The stool sample test is usually used as a screening tool for colon cancer. Do your results indicate that the stool sample test is still an effective screening tool?

We performed stool and mucosal sampling to identify differentially abundant taxa in stool and tumor samples in order to identify differences between these locations in regard to resident and transient flora. As seen in S2 e.g. Flavonifractor and Odoribacter were found in stool samples. Prior studies have reported the association of Flavonifractor in young-onset (<50 years) of CRC (DOI.org/10.1038/s41467-021-27112-y) and in alcoholic hepatitis patients it may induce oxidative stress and systematic inflammation (DOI:10.1080/19490976.2020.1785251.) Specific Odoribacter spp. were suggested to promote carcinogenesis (DOI.org/10.1038/s41598-022-14203-z).

Identifying specific microbial strains for screening is an important tool, but still, contrary to our publication on pancreatic cancer (10.1053/j.gastro2022.03.054; 10.1136/gutjnl-2021-324755), a signature that unifies the CRC microbiome across multiple studies has not been identified. Genera associated with CRC belong to Fusobacterium, Peptostreptococcus, Porphyromonas; Bacteroides, and Streptococcus. As mentioned our study looks at colon cancer from several sides (stool, tumor samples versus healthy small bowel and large bowel tissue, right versus left, preoperative versus postoperative) and aims to develop a deeper understanding of colon cancer.

  1. Lines 97 and 98. You mentioned, “most microbiome studies in CRC have focused on the analysis of fecal rather than tumor or mucosal samples”.  It is easier to obtain fecal samples than tumor or mucosal samples, and it is more practical to test fecal samples than mucosal samples. From this viewpoint, studying fecal samples may be more useful than tumor or mucosal samples. It is not clear how this study can be applied to diagnose or prevent colon cancer.

As mentioned our prospective study has looked at colon cancer from several sides (stool, tumor samples versus healthy small bowel and large bowel tissue, right versus left, preoperative versus postoperative) and aims to develop a deeper understanding of colon cancer, in particular in combination with detailed host factors. Furthermore, we believe that for understanding CRC heterogeneity it is important to regard CC and RC as two different tumor entities (p3, 98ff).  We are presenting one of the first studies dealing with a detailed analysis of different samples, including the terminal ileum.  It is necessary to understand the different aspects of the microbiome and the contribution to cancer development. Furthermore, our study analyzes postoperative changes in the stool microbiome. However, stool does not capture all the microbes in the gut, in particular microbes in the small intestine and mucosally adherent microbes (added p20, 590ff). Thus, we think stool samples are one aspect of the microbiome puzzle and tumor as well as healthy tissue samples a long with detailed patient information are another important aspect (p 21 630ff).  

  1. The quality of Figure 2 needs improvement. In addition, the notations (a) and (b) in Figure 2 should be put on top of the figures.

We thank the reviewer for pointing this out. The separately uploaded figure has a good resolution. We have amended the notations (a, b…) in figure 2 and all other figures as suggested.

Round 2

Reviewer 3 Report

I do not have other comments.